# Spatiotemporal observation of quantum crystallization of electrons

Hideaki Murase[1], Shunto Arai [1,2], Tatsuo Hasegawa [1], Kazuya Miyagawa [1] & Kazushi Kanoda [1,3,4,5] ✉

Liquids crystallize as they cool; however, when crystallization is avoided in some way, they supercool, maintaining their liquidity, and freezing into glass at low temperatures, as ubiquitously observed. These metastable states crystallize over time through the classical dynamics of nucleation and growth. However, it was recently found that Coulomb interacting electrons on charge-frustrated triangular lattices exhibit supercooled liquid and glass with quantum nature and they crystallize, raising fundamental issues: what features are universal to crystallization at large and specific to that of quantum systems? Here, we report our experimental challenges that address this issue through the spatiotemporal observation of electronic crystallization in an organic material. With Raman microspectroscopy, we have successfully performed real-space and real-time imaging of electronic crystallization. The results directly capture strongly temperature-dependent crystallization profiles indicating that nucleation and growth proceed at distinctive temperature-dependent rates, which is common to conventional crystallization. However, the growth rate is many orders of magnitude larger than that in the conventional case. The temperature characteristics of nucleation and growth are universal, whereas unusually fast growth kinetics features quantum crystallization where a quantum-to-classical catastrophe occurs in interacting electrons.

Crystallization from supercooled liquids or glasses[1–3], one of the most fundamental nonequilibrium phenomena, is widely observed in a range of condensed matter[2,4,5]. Conventionally, the phenomenon is governed by the classical dynamics of the constituent elements and is known to proceed via two sequential processes, the nucleation of crystal seeds and their growth. If the constituent elements have a quantum nature, how does the quantum nature affect the crystallization process that has ever been captured by classical dynamics and thermodynamics? The recent discovery of electron glass[6,7] and its crystallization[8] in molecular materials has offered an experimental avenue to tackle this fundamental question in soft matter physics.

In general, Coulomb interacting electrons tend to form a crystal called the Wigner crystal. In a real material, such an electronic crystal occurs as a charge order (CO) on the underlying lattice, which generally does not match the Wigner crystal in a continuum background so that the electrons crystalize so as to fit into the underlying lattice. When the lattice mismatch is significant, however, the electronic crystal can no longer conform to the material lattice while keeping regular periodicity but may form a glass, which is called charge glass (CG)[9–11]. For example, when electrons occupy a triangular lattice with an electron per two sites on average (a quarter-filled electronic band), they have difficulty in finding a stable CO owing to a large number of

[1]Department of Applied Physics, University of Tokyo, Bunkyo-ku, Tokyo 113-8656, Japan. [2]Research Center for Macromolecules and Biomaterials, National Institute for Materials Science (NIMS), Tsukuba, Ibaraki 305-0044, Japan. [3]Max Planck Institute for Solid State Research, Heisenbergstrasse 1, 70569 Stuttgart, Germany. [4]Physics Institute, University of Stuttgart, Pfaffenwaldring 57, 70569 Stuttgart, Germany. [5]Department of Advanced Materials Science, University of Tokyo, Kashiwanoha 5-1-5, Kashiwa 277-8561, Japan. ✉e-mail: k.kanoda@fkf.mpg.de

degenerate charge configurations; thus, they instead form CG[12–14]. Actually, the competitive appearance of CO and CG is demonstrated in the layered molecular conductors, θ-(BEDT-TTF)$_2$X [X = RbZn(SCN)$_4$, CsZn(SCN)$_4$ and TlCo(SCN)$_4$], with quasi-triangular lattices of BEDT-TTF molecules (Fig. 1a, b), where either CO or CG appears depending on the anisotropy of the triangular lattice controlling the degree of charge frustration[6,15,16]. The CG shows the hallmarks of conventional classical glasses, such as slow fluctuations, medium-scale correlation, and aging indicative of nonequilibrium[6,15,16]. Moreover, remarkably, the electrons in CG are found to have itinerant character by nuclear

magnetic resonance (NMR) and transport measurements, suggesting that CG has a quantum-classical energetic hierarchy[7].

Among these compounds, θ-(BEDT-TTF)$_2$RbZn(SCN)$_4$ (θ-RbZn) is a key material staying on the verge between CO and CG[17]. θ-RbZn undergoes a phase transition to CO at $T_{CO}$ = 200 K when cooled slowly, e.g., at a rate of 1 K/min[18–20]. However, when it is rapidly cooled at 5 K/min or faster, CO gives way to a supercooled charge liquid (SCL) and SCL freezes to CG at low temperatures (Fig. 1c)[6,15,16]. The crystallization from SCL or CG to CO was previously investigated by transport and NMR measurements, which characterized the evolution of the

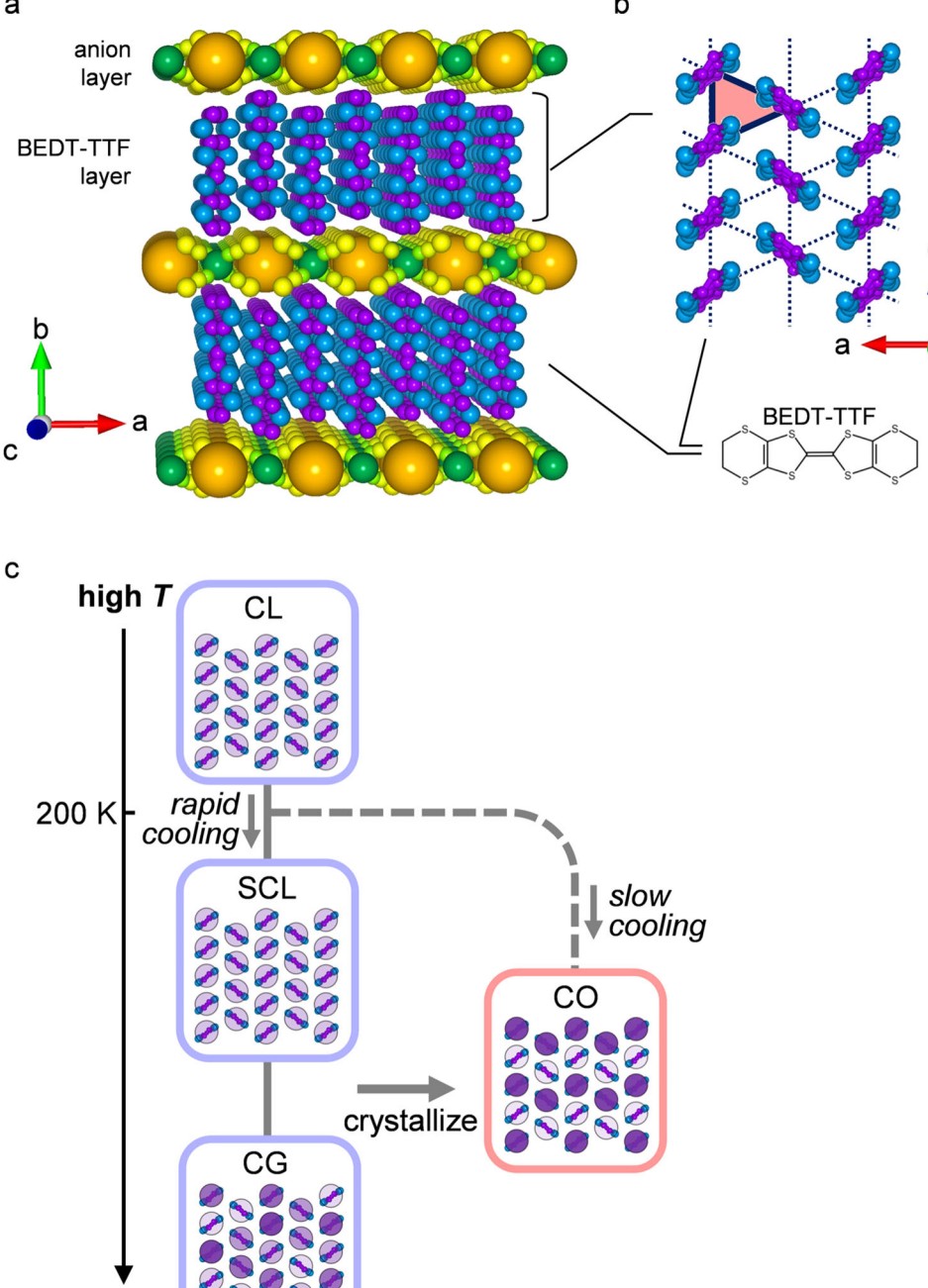

**Fig. 1 | Crystal structure and electronic phases of θ-(BDET-TTF)$_2$RbZn(SCN)$_4$.**
**a** Layered crystal structure of θ-(BEDT-TTF)$_2$RbZn(SCN)$_4$ viewed from the *c*-axis. The conducting BEDT-TTF layers are alternated with the insulating RbZn(SCN)$_4$ layers. The BEDT-TTF layers host charge order, charge glass, etc. **b** In-plane structure of the BEDT-TTF layer viewed from the *b*-axis. The BEDT-TTF molecules form

an anisotropic triangular lattice (highlighted by red color) with one hole per two molecular sites. **c** Schematic diagram of the temperature and cooling-rate dependence of the electronic phases in θ-(BEDT-TTF)$_2$RbZn(SCN)$_4$. CO, CL, SCL, and CG represent the charge order, charge liquid, supercooled charge liquid, and charge glass, respectively.

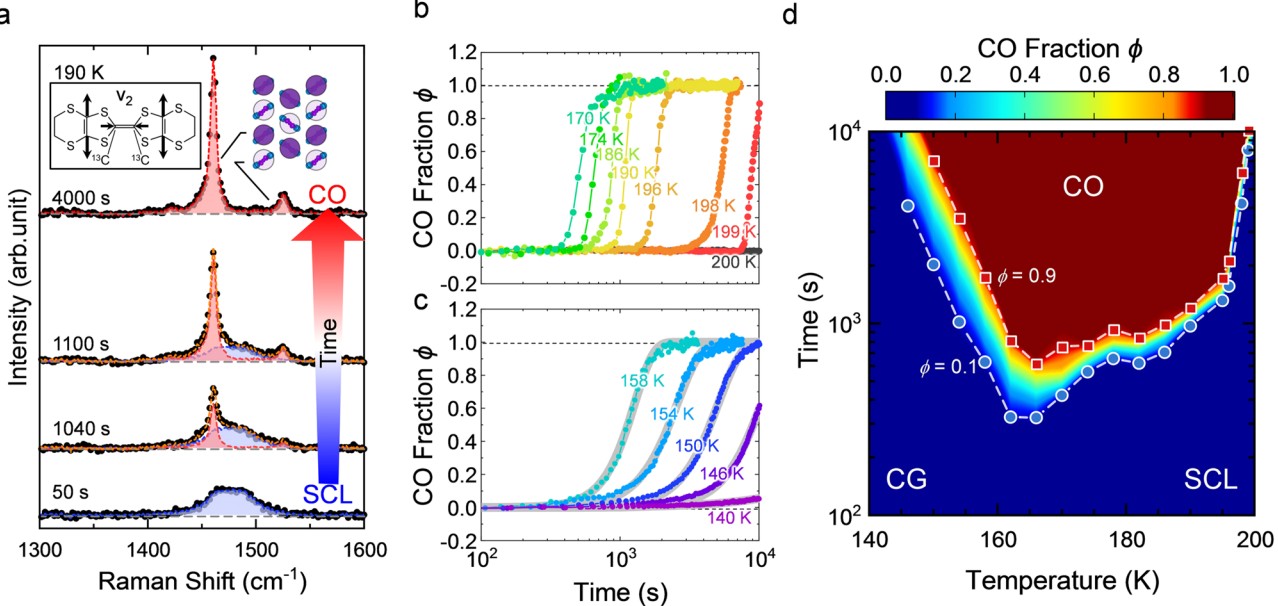

**Fig. 2 | Time evolution of the Raman spectra and CO fraction during electronic crystallization and the time–temperature-transformation (TTT) diagram.**
**a** Typical time evolution of the Raman spectra of the $v_2$ mode during the change from SCL/CG to CO. The excitation light polarized parallel to the $a$-axis is irradiated in a direction normal to the (010) surface, and the scattered light polarized parallel to the $a$-axis and $c$-axis is collected. Immediately after rapid cooling, the Raman spectrum of SCL/CG (blue) is observed. Then, at a certain time, the Raman peaks of CO (red component) emerge, and its fraction gradually increases to finally reach 100%. Inset: Charge sensitive $v_2$ mode of the BEDT-TTF molecule with the central double-bonded carbons enriched by $^{13}C$ isotopes. **b, c** Time evolution of the CO volume fraction $\phi$ at temperatures above (**b**) and below (**c**) the nose temperature. The gray lines are the fits of the Avrami equation (Eq. (3)) to the experimental points. **d** Contour plot of the $\phi$ values in the time–temperature-transformation plane (TTT diagram). The circles and squares indicate the contour points of $\phi = 0.1$ and 0.9, respectively; the dashed lines are guides for the eye. In the blue and brown areas, the sample is entirely occupied by CO and CG/SCL, respectively.

CO volume fraction and suggested different crystallization profiles at high and low temperatures; these results were reminiscent of classical glasses[8,21]. Spatiotemporal observation of crystallization is expected to directly reveal the nucleation and growth process to possibly elucidate the quantum effect; however, this has never been done. Here, we exploit Raman microspectroscopy to visualize the electron crystallization process in real space and real-time for the first time. The microscopic crystallization profile and the unusually rapid growth of crystal seeds not compatible with classical glass are presented.

## Results

Raman spectroscopy sensitively probes the molecular charge in organic conductors[22] by activating the C=C stretching modes in BEDT-TTF, which are known to be particularly charge-sensitive[23–26]. In the present study, we used the samples with the central double-bonded carbons enriched by $^{13}C$ isotopes (Fig. 2a), which makes the stretching mode well separated from other modes[19]. The Raman spectrum of this mode distinguishes between CO and SCL/CG. Figure 2a shows the time evolution of the Raman spectra of the charge-sensitive C=C stretching mode ($v_2$ mode) measured in an area of ~0.1 × 0.1 mm² after rapid cooling to 190 K. Just after that, an SCL is metastabilized so that the spectrum exhibits a broad peak indicative of a structureless nonuniform charge distribution. The spectral shape in SCL/CG is gradually and slightly temperature-dependent Supplementary Fig. 1). Then, it evolves into the two-peak structure of CO on the time scale of ~10³ s, as seen in Fig. 2a. In the CO phase, the Raman spectrum has two peaks coming from charge-rich and charge-poor molecules, reproducing the previous results[11]; the different peak intensities originate from the different Raman tensors of charge-rich and charge-poor sites[27], possibly related to electron–molecular vibration coupling.

Thus, the conversion from CG/SCL to CO is associated with not only charge rearrangement but also a change in charge density profile, distinct from the conventional crystallization of atoms, molecules, and colloids. In particular, the continuous charge density distribution in the CG/SCL state (Fig. 2a and Supplementary Fig. 1) is a remarkable feature of the electronic glass former. We note that the time scale of Raman spectroscopy, ~45 THz in the present case, is much faster than that of fluctuations responsible for glass formation (slower than 10 kHz) and even faster than thermal fluctuations (1.6–4.2 THz for 80–200 K); so, the observed spectrum projects the snapshot of the inhomogeneous charge distribution not motionally narrowed. Then, the continuous charge density distribution in CG/SC state is argued to come from the frustration-enhanced quantum-mechanical configuration interaction of various charge patterns and the temperature variation of the distribution width with no clear distinction between CG and SCL states is not due to classical motional narrowing but to thermally activated charge configurations that depend on temperature[28]. Indeed, very recent theoretical work[29] succeeded in explaining these properties of charge glass through quantum simulations.

The time evolution of the electronic crystallization is characterized by fitting the observed spectrum, $I(\nu, t)$, at a time, $t$, by a linear combination of the spectra of CO, $I^{CO}(\nu)$, and SCL/CG, $I^{CG}(\nu)$:

$$I(\nu, t) = A^{CO}(t)I^{CO}(\nu) + A^{CG}(t)I^{CG}(\nu), \qquad (1)$$

where $\nu$ is the Raman shift, and $A^{CO}$ and $A^{CG}$ are the fitting parameters reflecting the spectral weights (refer to Supplementary Note 1). Throughout crystallization, the experimental $I(\nu, t)$ is well-fitted by $I^{CO}(\nu)$ and $I^{CG}(\nu)$, and the volume fraction of CO is

$$\phi(t) = \frac{A^{CO}(t)}{A^{CO}(t) + A^{CG}(t)}. \qquad (2)$$

To globally characterize the temperature profile of CO evolution from SCL or CG, we examined the time evolution of $\phi(t)$ from the integrated Raman spectra over the areal scale of ~0.1 × 0.1 mm² during

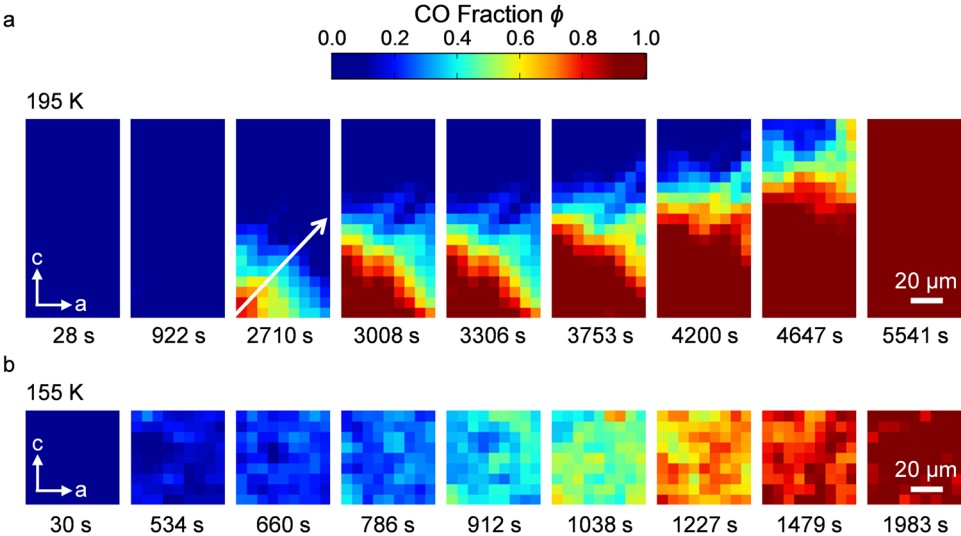

**Fig. 3 | Raman images of electronic crystal growth. a** Time evolution of the real-space Raman images (contour plots of $\phi$ on the sample ac plane) at 195 K. The images shown are those taken immediately after quenching to 195 K and at 922, 2710, 3008, 3306, 3753, 4200, 4647, and 5541 s for the starting time of the imaging. During this imaging, crystallization proceeds by ~6 μm, which is roughly equal to the spatial resolution of 6.5 μm (see Supplementary Note IV). The white arrow in the third image is the path, along which the growth rate is determined as shown in Fig. 4a. **b** Time evolution of the real-space Raman images at 155 K. The images shown are those taken immediately after quenching to 155 K and at 534, 660, 786, 912, 1038, 1227, 1479, and 1983 s. We took more images with narrower time intervals at 195 and 155 K (see Supplementary Note II for the images at other times). The size of one pixel is $6.5 \times 6.5$ μm², which is the spatial resolution of the present Raman imaging instrument.

isothermal crystallization at various temperatures. The previous XRD measurements, which revealed the temperature variation of correlation length, suggest that SCL crosses over or transitions to CG around 165 K upon cooling. The sample that was initially kept at 210 K, which is above $T_{CO}$, was rapidly cooled at 30 K/min to $T_q$, at which the Raman spectra were recorded as a function of time. After that, the sample was warmed to 210 K, and the sequence was repeated for different $T_q$ values. Figure 2b and c show the time evolution of $\phi(t)$ at $T_q$ values above and below 165 K, respectively. When $T_q$ is above 165 K, $\phi(t)$ vanishes for a while ($4 \times 10^2$–$10^4$ s) and then increases rapidly once it starts to rise (Fig. 2b). This means that the nucleation of the crystal seeds requires a certain incubation time, which is shorter at lower $T_q$, and immediately after nucleation occurs, the seeds spatially extend. In contrast, for $T_q$ below 165 K, the rise of $\phi$ from zero conversely requires a longer time at lower $T_q$ and is not sharp, followed by a gradual increase over time (Fig. 2c). In this temperature range, the time evolution of $\phi(t)$ is well described by the Avrami equation[30]:

$$\phi(t) = 1 - \exp\left(-Kt^n\right), \qquad (3)$$

where $K$ is the rate constant and $n$ is the Avrami exponent. This means that crystallization proceeds via the nucleation of numerous microcrystals and their growth. The deduced Avrami exponent $n$ is in the range of 2.6–3.5, which suggests that crystal growth proceeds in two-dimensional planes since $n$ is related to the spatial dimension of growth $d$ by $n = d + 1$ in ordinary cases. Figure 2d displays the contour plot of $\phi$ in the $t–T_q$ plane to construct the so-called time–temperature-transformation (TTT) diagram. The contour lines form nose structures characteristic of the nucleation and growth mechanism[2]. Remarkably, a nose temperature of ~165 K divides the high- and low-temperature regimes described above, consistent with previous transport and NMR studies[8,21].

To visualize the spatiotemporal evolution of the crystallization process, we utilized a Raman microspectroscopy technique with a spatial resolution of 6.5 μm, in essence, one pixel is $6.5 \times 6.5$ μm². Since the time evolution of $\phi$ behaves differently above and below $T_n$ (=165 K), we performed Raman microspectroscopy experiments at

$T_q = 195$ K ($>T_n$) and 155 K ($<T_n$) to comparatively examine the two regimes (Fig. 3a, b and Supplementary Note 2). Figure 3a shows snapshots of the crystallization in real space at $T_q = 195$ K, represented by the contour plot of $\phi$. The detailed time evolution of the snapshot (Raman image) is seen in Supplementary Movie 1. For a certain time after quenching, the system is an SCL everywhere in the sample (the blue color over the entire sample surface of $65 \times 130$ μm²). However, once a part of the SCL crystallizes to form a CO microdomain, it rapidly extends over the entire area before another nucleation occurs. Even when the observation area is extended to $124.8 \times 416$ μm², we observed only one nucleation that extends over the area (Supplementary Note 3), meaning that multiple nucleations are a rare stochastic event in the submillimeter scale. We confirm that the image does not practically evolve with time while taking one image in the present spatial and time resolutions (Supplementary Note 4). In the present experiments, the laser light penetrates the sample to a depth of about 1 μm which includes 500 layers (see the "Methods" section); so, the observed spectrum reflects an average of the charge states over this depth. The Raman image in Fig. 3a is roughly in blue or red with their boundary graded, meaning that the crystal growth from SCL proceeds in a three-dimensional manner with a possible layer-by-layer mixing of SCL and CO in the boundary region. In contrast, the contour plot of $\phi$ at $T_q = 155$ K shown in Fig. 3b indicates that the time evolution of $\phi$ occurs nearly homogeneously in the whole system (see also Supplementary Movie 2 for the detailed time evolution). This means that the spatial scale of nucleation and growth is considerably smaller than the resolution of 6.5 μm. In this case, the results are unable to distinguish whether the growth is two- or three-dimensional; however, the two-dimensional growth suggested by the Avrami analysis means that nucleated seeds extend with time within each layer to get mutually touched in the layers and may overlap incoherently out of the layers. The in-plane and out-of-plane coherence in the CO growing from CG is an intriguing issue to be investigated. A statistical error analysis shows that the mottled colors in Fig. 3b are not due to the spatial fluctuation of CO microcrystal density but to the spectral noise (Supplementary Note 5). Thus, the results shown in Fig. 3a and b demonstrate distinct spatial profiles of crystallizations from CG and SCL.

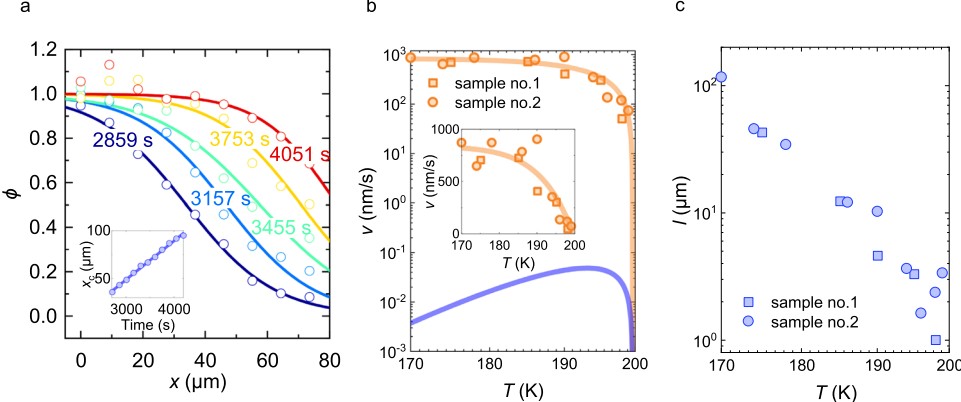

**Fig. 4 | Electronic crystal growth rate. a** $\phi$ values along the arrow in Fig. 3a at various times. The solid curves are fits of Eq. (4) to the data points. Inset: Time dependence of the location of the domain boundary $x_c$. **b** Temperature dependence of the experimental growth rate and theoretical curves. The blue line indicates the calculation based on the Wilson−Frenkel model. The orange line is a theoretical curve based on Eq. (5) with $k(T) = k_0$. **c** Temperature dependence of the growth step length $l$ projected by the Wilson−Frenkel model with the experimental values of $v$.

It is known that the times required for nucleation and the growth speed have different temperature dependencies. In the supercooled liquid state at high temperatures, nucleation takes a long time; however, the nuclei that happen to arise grow fast. In the glass state below $T_n$, nucleation events occur everywhere in the sample before individual nuclei spatially grow because of the slow growth speed; thus, crystallization proceeds in the form of an increasing number of fine nucleation sites, which are averaged and blurred at a spatial resolution of 6.5 μm. In such a case, the time evolution of the averaged volume fraction of the crystal seeds is known to be described by the Avrami equation[30], which is indeed followed by the experimental $\phi$ values (Fig. 2c).

Furthermore, we investigated the crystal growth speed by analyzing the time evolution of the Raman images. The $\phi$ values along the arrow at different times and a temperature of 195 K in Fig. 3a are plotted in Fig. 4a and fitted by the phenomenological form[31]:

$$\phi(x) = \frac{1}{2}\left\{1 + \tanh\left(\frac{x - x_c}{x_0}\right)\right\}, \qquad (4)$$

where $x_c$ and $x_0$ are fitting parameters and $x$ is the coordinate along the arrow. In this case, $x_c$, which represents the position of the SCL−CO interface, increases at a constant rate (the inset of Fig. 4a), suggesting that the interfacial reaction is the rate-controlling process. The slope of $x_c$ vs. $t$ defines the growth rate $v$, which is not appreciably dependent on the growth directions (Supplementary Note 6). We repeated these experiments and analyses at different temperatures to reveal the temperature dependence of $v$, the result of which is displayed in Fig. 4b. Notably, $v$ increases roughly linearly with $\Delta T = T_{CO}-T$ near $T_{CO}$ (the inset of Fig. 4b) and is saturated to a constant value of -10³ nm/s at low temperatures. In general, the temperature variation of $v$ is discussed as follows[32,33]:

$$v = k(T)\{1 - \exp(-\Delta\mu/k_B T)\}, \qquad (5)$$

where $k(T)$ is the temperature-dependent kinetic factor for crystal growth, $\Delta\mu = \mu_{SCL}-\mu_{CO}$ is the free energy difference between the SCL and CO, and $k_B$ is the Boltzmann constant. Notably, $\Delta\mu$ is proportional to $\Delta T$, which explains the $\Delta T$-linear variation of $v$ near $T_{CO}$.

Crystal growth in classical particle systems is generally described by the Wilson−Frenkel model[34], where $k(T)$ is proportional to the diffusion constant $D(T)$ in the supercooled liquid. We note that the diffusion constant in the Wilson−Frenkel model is defined for the motion that freezes to the glass state upon cooling whereas the diffusion

constant derived from the conductivity is for the transport of thermally excited single particles over a small charge gap and has no direct connection to the diffusion constant in question. Then, we used the characteristic frequency of the resistivity noise, $f(T)$, which probes the charge fluctuations responsible for SCL[6]. Then, Eq. (5) becomes:

$$v = f(T)l\{1 - \exp(-\Delta\mu/k_B T)\}, \qquad (6)$$

where $l$ is the length of one crystal growth step. As in the simplest case, we take a lattice constant of θ-RbZn (0.5 nm) as $l$ and exploit the previously obtained $f(T)$ by noise spectroscopy[6] (Supplementary Note 7). Since $\Delta\mu = \Delta H\Delta T/T_{CO}$ in which $\Delta H$ is the enthalpy difference between the SCL and CO (-160 K)[35], the substitution of the experimental values of $l, f(T)$ and $\Delta H$[35] into Eq. (6) gives the blue line in Fig. 4b; however, these values are 3–5 orders of magnitude below the experimental values. Note that this difference of several orders of magnitude at low temperatures is not affected by the choice of the value of $\Delta\mu$, namely $\Delta H$.

A possible modification of the Wilson−Frenkel model is to treat $l$ as an adjustable parameter. Figure 4c is the plot of $l = v[f(T)\{1 - \exp(-\frac{\Delta\mu}{k_B T})\}]^{-1}$ with the experimental values of $v, f(T)$, and $\Delta H$. It turns out that $l$ reaches an unrealistically large value of 100 μm (10⁵ lattice constants) at 170 K. It is noted that $l$ can be longer than the lattice constant in the avalanche-mediated crystallization of the mature glass[36]; however, it is at most on the order of 10 lattice constants. The $l$ may be related to the dynamical correlation length, which was suggested to increase toward the glass transition in colloidal systems[37]. The revealed dynamical correlation length at the glass transition is however several times the colloid spacing; the length scale is quite different from the present case in which $l$ is orders of magnitude larger than the electron spacing. Alternatively, one may assume that $k(T)$ is not proportional to a diffusion constant, as is the case in some classical particle systems, where $k(T)$ is gapless[38–41]. This is possible because of the crystalline-like local structure of the liquid abutting the crystal surface[32,33]. However, the fit with $k(T) = k_0$ (the orange line in Fig. 4b) yields $k_0 = 840$ nm/s and $\Delta H = 4600$ K, which is 30 times larger than the experimental value.

## Discussion

Thus, the experimental growth rates are not reconciled with the conventional models. This discrepancy is a key to the microscopic mechanism of the electronic crystallization to be clarified. As a possible relevance to the extraordinarily fast crystallization, we suggest the quantum nature of electrons inherited by the CG or SCL, which has

moderate electrical conductivity, e.g., 10 S/cm at 150 K, that is suggestive of a spatially spreading wave function[7] and contrasts to the highly insulating CO state with the wave function well localized on each molecule. In classical systems, crystallization proceeds by the spatial rearrangement of particles; however, crystallization from the CG or SCL should be accompanied by shrinking of the wave functions and a drastic change in the quantum nature of electrons, which results in a novel case of quantum crystallization. Kinetic factor $k(T)$ values that are orders of magnitude larger than those in the conventional case are known in the quantum crystallization of liquid He[42,43]. Electronic and He crystallizations share anomalous nonequilibrium dynamics.

We mention that the electronic crystallization should proceed with electronic configurational change and lattice distortion mutually coupled as a novel process never seen before in the conventional crystallization phenomena; in particular, the role of the lattice distortion in the crystallization is an open issue to be addressed in the future. Nevertheless, we can argue that the lattice may not regulate the crystallization speed, noting that the speed of sound, namely, that of long wavelength phonons is $10^9$ times faster than the crystallization speed. Although the short-wave-length phonon, which is generally slower than the sound velocity, is responsible for the transmission of the local lattice distortion, it should still be several orders of magnitude faster than the present crystallization speed. We also note that the numerical simulation of charge glass on a rigid lattice[29] reproduces the observed behavior of relaxation time[6,15], which is likely responsible for the crystallization speed.

In the present study, we succeeded in directly observing the spatiotemporal profile of electronic crystallization, evidencing distinct crystallization features at high and low temperatures that are consistent with the celebrated nucleation and growth mechanism. On the other hand, it has been found that the liquid (or glass) crystal interface proceeds through space several orders of magnitude faster than expected by the Wilson-Frenkel model relevant to classical systems. We speculate that the quantum nature of electrons is deeply involved in the anomalously high speed of crystal growth. Indeed a recent theory has shown that the quantum nature of electrons affects the physical properties of charge glass[29]. The electronic state at the SCL-CO interface and its dynamics should hold the key to the microscopic mechanism of fast growth. The present results are expected to open a new avenue to the physics of crystallization in quantum systems and trigger theoretical studies on this issue.

## Methods

The Raman spectrum of the C=C stretching mode ($\nu_2$ mode) in BEDT-TTF, which is particularly charge-sensitive[23–26], was exploited to distinguish between CO and SCL/CG, and their spatial distribution and time evolution were visualized by Raman microspectroscopy. We used $^{13}$C-enriched single crystals of θ-RbZn, which were synthesized by the electrochemical oxidation method because the assignment of Raman peaks was straightforward owing to resolving of the degeneracy of the C=C stretching modes by site-selective substitution[19]. The typical crystal size was approximately $1 \times 0.1 \times 0.1 \, mm^3$. The crystals were mounted on copper substrates and then loaded on a cooling stage (Linkam 10002 L), which had a glass window for optical observation. We cooled the sample at a rate of 30 K/min to metastabilize the SCL and CG; in other cases, cooling was performed at 1 K/min or more slowly. Raman spectra were measured by a Renishaw inVia Raman system. Excitation was provided from a 532 nm laser focused through a microscope equipped with a Leica N Plan L×50 objective lens. The scattered light in the backscattering geometry was split by a diffraction grating of 1800 g/mm and detected by a charge coupling device. The Renishaw StreamLine technique was used for rapid Raman imaging. The investigated temperature range was 140–200 K.

The observation depth is given by the smaller of the following two lengths. One is the depth resolution of the microscope. The formula of full width at half minimum for a confocal microscope is as follows:

$$0.64 \times \frac{\lambda}{n - \sqrt{n^2 - NA^2}},$$

where $\lambda$ is a wavelength (532 nm), $n$ is a refractive index (1) and NA is a numerical aperture of an objective lens (0.5). The substitution of these values yields a depth resolution of 2.5 μm. The other is the penetration depth of the light, given as the inverse of the absorption coefficient. According to S. Iwai et al.[44], the absorption coefficient of θ-RbZn is 15,000 cm$^{-1}$, which means that the penetration depth is 0.67 μm. Then, the observation depth turns out to be about 1 μm.

## Data availability

The data that support the discussion and conclusion in the present paper are all presented in the main manuscript and Supplementary Information online. Additional data, e.g., numerical values, are available from the corresponding author upon request.

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

## Acknowledgements

We thank T. Sato and K. Yamamoto for their initial trial of Raman spectroscopy on electronic crystallization, which motivated the present work, and A. Ikeda, T. Kato, and K. Yoshimi for their fruitful discussion. This work was supported by the Japan Society for the Promotion of Science (JSPS) under Grant Numbers 18H05225 (K.K.), 19H01846 (K.M.), 20K20894 (K.M.), 20KK0060 (K.M. and K.K.), 19H02579 (S.A.) and 21H05234 (S.A.). S.A. also thanks to the support from Murata Science Foundation.

## Author contributions

K.M. prepared samples. H.M. performed experiments and analyzed as well as interpreted data with the help of S.A., T.H., K.M., and K.K. K.K. designed the project. H.M. and K.K. wrote the manuscript with input from all authors.

## Funding

## Competing interests

The authors declare no competing interests.
