## [Peer Review File · Nature Communications]

REVIEWER COMMENTS

Reviewer #1 (Remarks to the Author):

This communication reports the real-space and real-time imaging of electronic crystallization from the charge glass (CG) and super cooled charge liquid (SCL) states in layered molecular conductors. The CG and SCL have attracted much attention as exotic electronic phases. Molecular conductors such as theta-BEDT-TTF compounds that have Coulomb interacting electrons have been a representative example showing such exotic electronic properties. The electron crystallization processes from the CG or SCL state to a state with charge order (CO) have been understood from the viewpoint of the macroscopic nucleation on the high-temperature side and the growth of the microcrystals (\ll spatial resolution) on the low-temperature side on the basis of the resistivity, NMR and XRD measurements. In this manuscript, the authors claim that their spatiotemporal measurements reflect the quantum nature of electron crystallization.

The authors distinguish the CO state from the SCL state by Raman microscopy to obtain the spatiotemporal image (resolution of 6 microns x 6 microns). They nicely observe how the crystallization is spatially developed. Interestingly, the spatial dynamics of the macroscopic nucleation on the high-temperature side are very different from those of the microscopic crystal growth on the low-temperature side. They obtain the temperature dependence of the velocity of the macroscopic nucleation from their experiment. Furthermore, the authors analyze them by using a phenomenological model for the crystal growth called the Wilson-Frenkel model, in which f (frequency of the fluctuation) $\times l$ (crystal growth step) is substituted for the diffusion constant $D(t)$.

The measurements are carefully and well organized, and the methods of analysis are quite reasonable. However, I am afraid that their conclusion is not supported by their analysis. They claim that the value of l in eq (6) is much larger (10^5 times) larger than the expected value (= lattice constant) in their analysis within this model. According to this fact accompanied with additional considerations, they conclude that their observations reflect the quantum nature of the electron crystallization. However, it does not rule out the possibilities that the above discrepancies of " l " may be understood by a different mechanism.

In eq. (6), they use the frequency of the fluctuation obtained from the noise spectroscopy. This frequency would inevitably become low because the observed fluctuations are limited by the macroscopic aspects of the fluctuation, even if electronic fluctuations exist. The authors should use a more macroscopic crystal growth step accounting for the macroscopic fluctuation (instead of the microscopic one they employ such as the lattice constant). The discrepancy of 10^5 might almost disappear, if they use a macroscopic step in their analysis. In this manuscript and its references, I could not find any justification for employing the microscopic growth step for the slow fluctuations. From this standpoint, the conclusion of this manuscript is not justified.

The temperature dependent electron crystallization has indeed been interesting. I agree that the results of the macroscopic Raman imaging in this manuscript are very important. (In particular, the

difference between the spatiotemporal pattern on the high-temperature side and that on the low-temperature side). However, I don't agree to the conclusion because of the above critical reason.

In summary, I agree that the results shown here include new information for understanding electron crystallization. However, their conclusion is not justified from their arguments. Therefore, I regretfully cannot recommend this manuscript for publication.

Reviewer #2 (Remarks to the Author):

The manuscript by H. Murase et al "Spatiotemporal observation of quantum crystallization of electrons" explores charge order/charge glass system to study electronic crystallization from overcool liquid. The authors quantitatively characterize this process both as a function of time and following the way electronic crystal forms in space. They used Raman scattering spectroscopy as a tool for the charge glass/charge crystal characterization. The temporary and spatial characteristics of the formation of a charge crystal from overcooled charge glass obtained this way are compared to the parameters of the formation of regular classical glasses. The authors find that the charge glass growth rate is many orders of magnitude faster than in a conventional glass case, while the temperature characteristics of nucleation and growth are universal. They assign it to the fact, that the glass is quantum (electronic) glass, where not diffusion coefficient of particles, but spread of electronic orbitals defines crystallisation processes.

I find the paper very interesting, and appealing to the audience working in fields of both soft and hard condensed matter. To my knowledge is it the first detailed comparison of regular glass vs electronic glass crystallization processes. The data and the analysis are robust and new. I think that the paper can be published with Nature Communications if the authors can provide answers to the questions and comments I present below.

1. For audience not familiar with charge-ordered BEDT-TTF organic materials it is not clear what is the charge order, what is the origin of it, what role does frustration of the lattice play in formation of these states? What is the amplitude of the charge order? Is the amplitude important? Is the charge separation amplitude different between charge glass and charge order, and why?

I can judge from Raman spectra that the charge separation is much larger in the CO state compared to CG/SCL. So on the electronic crystallization authors observe not only ordering of the charges A and B, but also increase of the charge separation amplitude between A and B, unless CG/SCL Raman spectra are defined by high fluctuation rate between the two charge states A and B. This is not discussed in the paper. The author need to comment on that, either to determining the charge fluctuations in CG/SCL state, or explaining how an increase of charge separation amplitude fits into the picture.

2. Statement in lines 42-43 I find unclear: "For example, on a triangular lattice that is half-filled (a band is quarter filled) by electrons, they have difficulty in finding a stable..." - What is half filled, and what is quarter filled?

3. Ref 16 typo in the reference: Phys. Rev. A 86, 023604 (2012). This paper describes spinless Fermions with random long range interactions. The authors refer to the lattice frustration as an origin of the spin glass state. Does randomness also play a role in the formation of the CG? If not, is the glassy state the authors discuss actually related to the case discussed in Ref. 16?

4. θ -RbZn undergoes a CO transition with a large lattice change when cooled down slowly. Is this the case for the charge order crystallized from the overcooled liquid, does this crystallization occurs with the distortion of the lattice? Does the lattice response matter for crystallization?

5. Can authors distinguish between charge glass and supercooled charge liquid? How? I see in the paper that these two states are ultimately distinguish by the dynamics of the crystallisation process. Is there any other distinction? The authors should comment on this.

6. The authors show that electronic crystal growth occurs in two dimensions, as suggested by the Avrami exponent. Does coupling between the BEDT-TTF planes matter? What is the penetration depth of the Raman probe? I assume it does not have a capacity to probe a single layer. Do the authors think that the charge order forms in 3D and is synchronized between the BEDT-TTF planes/ - I assume in any other case they would not observe such a distinct picture, because they would observe different overlapping responses from different layers within penetration depth. These responses would be asynchronized resulting in an averaged picture.

7. Figure 3 shows the snapshots of crystallization. How long does it take to obtain such a snapshot? Is the time shown on the figure a starting time? How does the time of the snapshot acquisition compare to the crystallization rate?

8. A question about the analysis of crystallization rate from Raman images. Does the length of the interface between SCL and CO influenced crystallisation? Fig. 3a shows only one seed in the Raman image. Is this picture an example of crystallization images? Do they have any statistics on the crystallization process? I do not see a reason why there should be only one seed in the window of the snapshot. The manuscript tells that it takes time to get the seeds of crystallization in the SCL state. Is the density of those seeds a random number?

9. The authors show a discrepancy of the growth rate of a crystal between a classical SCL and charge SCL they study. Their statement is based on the application of the Wilson-Frenkel model, where they use charge fluctuations rate instead of diffusion coefficient as is done in case of classical crystallization. It is not completely clear to me why the charge fluctuations rate is a good parameter to substitute for the diffusion coefficient. The authors mention earlier in the paper as well as after immediately after the discussion of Wilson-Frenkel model, that the glass is conducting. Therefore there is some charge carriers diffusion associated with the carriers mobility. Can this number be calculated and used as a diffusion coefficient? - it will basically give an qualitative estimate of "electronic orbitals spread" the authors mention. The authors should discuss how this influences crystallisation.

Reviewer #3 (Remarks to the Author):

The authors have performed Raman microspectroscopy to study crystallization dynamics of electrons in an organic glassy system θ -RbZn.

Raman microspectroscopy allows them to detect a distribution of charge occupancy of the molecular sites, thus distinguishing between an inhomogeneous distribution of charge and the stripe charge ordered phase.

To observe the relevant timescales, the sample was rapidly cooled to a "quench temperature" T_q after which the dynamics of charge ordering was measured. They observed a typical feature that nucleation goes faster as the temperature is lowered, up to a 'nose temperature' of $T=165$ K, below which the nucleation of the crystal becomes progressively slower.

By showing the spatial spread of the crystalline phase, they observe two distinct regimes: one with clear nucleation and fast domain growth; whilst at lower temperatures the crystalline nucleation is more homogeneous without domain growth.

At lower temperatures a comparison to the Wilson-Frenkel model fails to explain their data.

This, as the authors claim, is due to the “quantum” nature of the glassy freezing.

I think the data presented is really exciting and provides an almost unique and unprecedented insight into glassy slowdown of dynamics. However, the claim that the larger growth rate at low temperatures is attributable to “the quantum effect” is not supported by any clear theory. I can recommend publication of the manuscript, once the authors clarify this statement. That is: either substantiate the “quantum” claim or just stick with the observation that the growth rate violates Wilson-Frenkel theory.

In addition, I have a few other questions and one typo:

- One possible interpretation put forward in the discussion around Fig 4b is that there is a growing length scale “ l ” (fig 4c) responsible for the large growth rate. This might be an interesting point, and there is a lot of discussion in the glass literature that is not mentioned here, see for example “Direct Experimental Evidence of a Growing Length Scale Accompanying the Glass Transition”, Berthier et al, Science 310, 5755 (2005). I would appreciate if the claim of a growing length scale in Fig 4c is accompanied by an in-depth discussion of these ideas in related glass-problems.
- Is there a distinction in Raman signal between the high-temperature phase (electron liquid) and the frozen electron charge glass at low T? Because in both cases, the charge distribution on a single site has a broad distribution, but in the high T case its fluctuating whereas at low T it’s freezing.
- Typo, in line 114 and 192 there is written “spaciotemporal” instead of spatiotemporal.

Replies to Reviewers and descriptions of revisions

[Replies to Reviewer #1]

[Comment #1-0]

This communication reports the real-space and real-time imaging of electronic crystallization from the charge glass (CG) and super cooled charge liquid (SCL) states in layered molecular conductors. The CG and SCL have attracted much attention as exotic electronic phases. Molecular conductors such as theta-BEDT-TTF compounds that have Coulomb interacting electrons have been a representative example showing such exotic electronic properties. The electron crystallization processes from the CG or SCL state to a state with charge order (CO) have been understood from the viewpoint of the macroscopic nucleation on the high-temperature side and the growth of the microcrystals (\ll spatial resolution) on the low-temperature side on the basis of the resistivity, NMR and XRD measurements. In this manuscript, the authors claims that their spatiotemporal measurements reflect the quantum nature of electron crystallization. The authors distinguish the CO state from the SCL state by Raman microscopy to obtain the spatiotemporal image (resolution of 6 microns x 6 microns). They nicely observe how the crystallization is spatially developed. Interestingly, the spatial dynamics of the macroscopic nucleation on the high-temperature side are very different from those of the microscopic crystal growth on the low-temperature side. They obtain the temperature dependence of the velocity of the macroscopic nucleation from their experiment. Furthermore, the authors analyze them by using a phenomenological model for the crystal growth called the Wilson-Frenkel model, in which f (frequency of the fluctuation) $\times l$ (crystal growth step) is substituted for the diffusion constant $D(t)$.

[Reply #1-0]

We are grateful to Reviewer #1 for reading our manuscript thoroughly and acknowledging the significance of the present work. Below, we reply to the critiques raised by the reviewer.

[Comment #1-1]

The measurements are carefully and well organized, and the methods of analysis are quite reasonable. However, I am afraid that their conclusion is not supported by their analysis. They claim that the value of l in eq (6) is much larger (10^5 times) larger than the expected value (= lattice constant) in their analysis within this model. According to this fact accompanied with additional considerations, they conclude that their observations reflect the quantum nature of the electron crystallization. However, it does not rule out the possibilities that the above discrepancies of “ l ” may be understood by a different mechanism.

In eq. (6), they use the frequency of the fluctuation obtained from the noise spectroscopy. This frequency would inevitably become low because the observed fluctuations are

limited by the macroscopic aspects of the fluctuation, even if electronic fluctuations exist. The authors should use a more macroscopic crystal growth step accounting for the macroscopic fluctuation (instead of the microscopic one they employ such as the lattice constant). The discrepancy of 10^5 might almost disappear, if they use a macroscopic step in their analysis. In this manuscript and its references, I could not find any justification for employing the microscopic growth step for the slow fluctuations. From this standpoint, the conclusion of this manuscript is not justified.

The temperature dependent electron crystallization has indeed been interesting. I agree that the results of the macroscopic Raman imaging in this manuscript are very important. (In particular, the difference between the spatiotemporal pattern on the high-temperature side and that on the low-temperature side). However, I don't agree to the conclusion because of the above critical reason.

[Reply #1-1]

The Wilson-Frenkel model is widely applied to the conventional crystallization of supercooled liquid (SCL). This model assumes that the crystallization proceeds step by step with an elementary event that occurs at the frequency, f , of the so-called α -relaxation responsible for the dynamics of SCL and in the length scale, l , of the order of interparticle distance (or lattice constant). In the conventional Wilson-Frenkel model, there is no reason why the interparticle spacing is replaced by any macroscopic length that is several orders of magnitude larger than that although possibly extended by a factor. In conventional classical particle systems, the f is known from the diffusion constant or viscosity. In the present system, f is directly known from the resistivity fluctuations and aging behavior. [Please see also Reply #2-9.] Thus deduced f values were confirmed to be those of the SCL dynamics in that the f values in the resistivity fluctuations and aging behavior are on a single activation function and the glass transition temperature suggested by the temperature dependence of the f values nearly coincide with the practical glass transition observed, e.g., by X-ray diffraction (Refs. 6, 15). Thus, if one considers the present crystallization process in terms of the conventional classical scheme, one should adopt these experimentally obtained values as f . If orders of magnitude faster fluctuations or orders of magnitude longer step-lengths are required to explain the observation, it means the breakdown of the classical Wilson-Frenkel scheme of crystallization. In short, the present observation of the crystallization speed necessitates orders of magnitude faster fluctuations and/or orders of magnitude longer step than expected in the classical Wilson-Frenkel scheme, indicating the violation of the classical scheme.

[Comment #1-2]

In summary, I agree that the results shown here include new information for understanding electron crystallization. However, their conclusion is not justified from their arguments. Therefore, I regretfully cannot recommend this manuscript for publication.

[Reply #1-2]

As we mentioned above, we believe that our arguments make sense and hope our reply is convincing to the reviewer.

[Replies to Reviewer #2]

[Comment #2-0]

The manuscript by H. Murase et al "Spatiotemporal observation of quantum crystallization of electrons" explores charge order/charge glass system to study electronic crystallization from overcool liquid. The authors quantitatively characterize this process both as a function of time and following the way electronic crystal forms in space. They used Raman scattering spectroscopy as a tool for the charge glass/charge crystal characterization. The temporary and spatial characteristics of the formation of a charge crystal from overcooled charge glass obtained this way are compared to the parameters of the formation of regular classical glasses. The authors find that the charge glass growth rate is many orders of magnitude faster than in a conventional glass case, while the temperature characteristics of nucleation and growth are universal. They assign it to the fact, that the glass is quantum (electronic) glass, where not diffusion coefficient of particles, but spread of electronic orbitals defines crystallisation processes.

I find the paper very interesting, and appealing to the audience working in fields of both soft and hard condensed matter. To my knowledge is it the first detailed comparison of regular glass vs electronic glass crystallization processes. The data and the analysis are robust and new. I think that the paper can be published with Nature Communications if the authors can provide answers to the questions and comments I present below.

[Reply #2-0]

We are grateful to Reviewer #2 for reading our manuscript thoroughly and acknowledging the significance of the present work. Having enlightening comments, we scrutinized the previous MS and then revised it. Below, we reply to all of the comments and describe the revised parts, which are underlined in the following replies and highlighted by red characters in the MS.

[Comment #2-1]

1. For audience not familiar with charge-ordered BEDT-TTF organic materials it is not clear what is the charge order, what is the origin of it, what role does frustration of the lattice play in formation of these states? What is the amplitude of the charge order? Is the amplitude important? Is the charge separation amplitude different between charge glass and charge order, and why?

I can judge from Raman spectra that the charge separation is much larger in the CO state compared to CG/SCL. So on the electronic crystallization authors observe not only ordering of the charges A and B, but also increase of the charge separation amplitude between A and B, unless CG/SCL Raman spectra are defined by high fluctuation rate between the two charge states A and B. This is not discussed in the paper. The author need to comment on that, either to determining the charge fluctuations in CG/SCL state,

or explaining how an increase of charge separation amplitude fits into the picture.

[Reply #2-1]

Following the reviewer's suggestions, we revised the manuscript as follows.

In reply to "what is the charge order, what is the origin of it, what role does frustration of the lattice play in the formation of these states?", we added the following explanations in the revised MS:

[3rd paragraph] "In general, Coulomb interacting electrons tend to form a crystal called the Wigner crystal. In a real material, such an electronic crystal occurs as a charge order (CO) on the underlying lattice, which generally does not match the Wigner crystal in a continuum background so that the electrons crystalize so as to fit to the underlying lattice. When the lattice mismatch is significant, however, the electronic crystal can no longer conform to the material lattice with keeping regular periodicity but may form a glass, which is called charge glass (CG)⁹⁻¹¹."

As the reviewer pointed out, the conversion from CG/SCL to CO is associated with not only charge rearrangement but also a change in charge density profile ("amplitude" in the reviewer's words). This is a distinct point of electronic crystallization not seen in the conventional crystallization of atoms, molecules, and colloids. As Fig. 2a and Fig. S1 suggest, the charge density distribution is double-peaked in the CO state, as expected, but is continuous in the CG/SCL state. As the latter is a characteristic property of charge glass, we investigated, for these three years, the material and temperature dependences of the charge density profiles, which are deduced from the Raman spectra, for a series of materials, θ -(BEDT-TTF)₂X (X=RbZn(SCN)₄, TiCo(SCN)₄, CsZn(SCN)₄, and I₃), with triangular lattices of different anisotropy, namely, different degrees of charge frustration. The results are reported in Murase et al., (arXiv: 2205.10795). What we observed is that the charge-density distribution in the CG state gets narrower as the lattice frustration increases and the distribution shape somewhat changes with temperature. We note that the time scale of Raman spectroscopy (45 THz in the present case) is much faster than fluctuations responsible for glass formation (slower than 10 kHz below room temperature) and even faster than thermal fluctuations (1.6-4.2 THz for 80-200 K); so, the observed spectrum reflects the snapshot (not motionally narrowed) of the inhomogeneous charge distribution. Then, we argued that the continuous and narrowed charge density distribution in the CG state comes from the frustration-enhanced quantum-mechanical configuration interaction of various charge patterns and the temperature variation of the distribution width is ascribable to thermally activated charge configurations that depend on temperature. Very recent theoretical work by S. Fratini et al. (arXiv:2208.06260) succeeded in explaining these properties of charge glass in terms of quantum simulations. In the revised MS, we have added the following paragraph for the explanation relevant to the scope of the present work by referring to Murase et al., (arXiv: 2205.10795) for more detail:

[6th paragraph] "Thus, the conversion from CG/SCL to CO is associated with not only charge rearrangement but also a change in charge density profile, distinct from the conventional crystallization of atoms, molecules, and colloids. In particular, the continuous charge density distribution in CG/SCL state (Fig. 2a and Fig. S1) is a

remarkable feature of the electronic glassformer. We note that the time scale of Raman spectroscopy, ~ 45 THz in the present case, is much faster than that of fluctuations responsible for glass formation (slower than 10 kHz) and even faster than thermal fluctuations (1.6-4.2 THz for 80-200 K); so, the observed spectrum projects the snapshot of the inhomogeneous charge distribution not motionally narrowed. Then, the continuous charge density distribution in CG/SC state is argued to come from the frustration-enhanced quantum-mechanical configuration interaction of various charge patterns and the temperature variation of the distribution width with no clear distinction between CG and SCL states is not due to classical motional narrowing but to thermally activated charge configurations that depend on temperature [28]. Indeed, very recent theoretical work [29] succeeded in explaining these properties of charge glass through quantum simulations.”

[28] Murase et al., Observation of classical to quantum crossover in electron glass. arXiv:2205.10795

[29] Fratini et al. A quantum theory of the nearly frozen charge glass. arXiv:2208.06260

[Comment #2-2]

2. Statement in lines 42-43 I find unclear: "For example, on a triangular lattice that is half-filled (a band is quarter filled) by electrons, they have difficulty in finding a stable..."
- What is half filled, and what is quarter filled?

[Reply #2-2]

The half-filling is the electron occupation per site, while the quarter-filling is in the sense of an electronic band. To state this more clearly, we modified the relevant descriptions as follows;

[3rd paragraph] *“For example, when electrons occupy a triangular lattice with an electron per two sites on average (a quarter-filled electronic band)”*

[Comment #2-3]

3. Ref 16 typo in the reference: Phys. Rev. A 86, 023604 (2012). This paper describes spinless Fermions with random long range interactions. The authors refer to the lattice frustration as an origin of the spin glass state. Does randomness also play a role in the formation of the CG? If not, is the glassy state the authors discuss actually related to the case discussed in Ref. 16?

[Reply #2-3]

Although we cited this paper in relevance to fermion glass as in the present case, citing it may be misleading since the randomness-driven glass formation in it is different from the present frustration-driven and disorder-free mechanism, as the reviewer

suggested. So, we deleted this reference.

[Comment #2-4]

4. Υ theta-RbZn undergoes a CO transition with a large lattice change when cooled down slowly. Is this the case for the charge order crystalized from the overcooled liquid, does this crystallization occurs with the distortion of the lattice? Does the lattice response matter for crystallization?

[Reply #2-4]

As the reviewer suggested, the transition from SCL or CG to CO should be accompanied by a sizable lattice deformation. The electronic crystallization proceeds with the electron rearrangement with charge density modulation and the lattice distortion mutually coupled. This is an unknown and novel process not observed in the conventional crystallization phenomena so far studied; in particular, the role of the lattice distortion in the crystallization is an open issue to be addressed in the future. Nevertheless, we can argue that the lattice may play only an auxiliary role in the evolution of the crystallization by comparing the observed crystallization speed and the phonon velocity. The former is 10^9 times slower than the sound velocity, which characterized the long wavelength phonons. Although the transmission of the local lattice distortion is determined by the short-wave-length phonon and is generally slower than the sound velocity, the present crystallization speed should keep orders of magnitudes slower than the phonon velocity. In addition, the recent numerical simulation of charge glass on a rigid lattice [Ref.29] reproduces the relaxation-time behavior of charge glass [Refs. 6 and 15], which is believed to determine the crystallization speed. To mention the above, we have added the following explanation to the revised MS:

[15th paragraph] “We mention that the electronic crystallization should proceed with electronic configurational change and lattice distortion mutually coupled as a novel process never seen before in the conventional crystallization phenomena; in particular, the role of the lattice distortion in the crystallization is an open issue to be addressed in the future. Nevertheless, we can argue that the lattice may not regulate the crystallization speed, noting that the speed of sound, namely, that of long wavelength phonons is 10^9 times faster than the crystallization speed. Although the short-wave-length phonon, which is generally slower than the sound velocity, is responsible for the transmission of the local lattice distortion, it should still be several orders of magnitude faster than the present crystallization speed. We also note that the numerical simulation of charge glass on a rigid lattice [Ref.29] reproduces the observed behavior of relaxation time [Refs.6, 15], which is likely responsible for the crystallization speed”

[Comment #2-5]

5. Can authors distinguish between charge glass and supercooled charge liquid? How? I see in the paper that these two states are ultimately distinguish by the dynamics of the

crystallisation process. Is there any other distinction? The authors should comment on this.

[Reply #2-5]

The charge glass is a nonequilibrium state whereas the supercooled charge liquid is an equilibrium state. The Raman spectrum reflects the snapshot of the charge density profiles (please see Reply #2-1) and is unable to distinguish the two states concerning dynamics. In general, glass and supercooled liquid are often not sharply distinguishable, but differences can appear in some physical behaviors. In the CG/SCL case, it is the presence or absence of the nonequilibrium aging behavior of resistivity (Ref.15) and the leveling-off or growth of the correlation length detected by XRD upon cooling (Refs. 6). For the present system, θ -(BEDT-TTF)₂RbZn(SCN)₄, the XRD suggests that CG and SCL reside below and above 160-170 K (Ref. 6). It corresponds to the nose temperature in the TTT curve (Fig. 2d), and the present results (Figs. 3a and 3b) appear to distinguish CG and SCL in their different spatial profiles of crystallization. In the revised MS, we added the following explanations:

[8th paragraph] “The previous XRD measurements, which revealed the temperature variation of correlation length, suggest that SCL crosses over or transitions to CG around 165 K upon cooling.”

and

[9th paragraph] “Thus, the results shown in Figs. 3a and 3b demonstrate distinct spatial profiles of crystallizations from CG and SCL.”

[Comment #2-6]

6. The authors show that electronic crystal growth occurs in two dimensions, as suggested by the Avrami exponent. Does coupling between the BEDT-TTF planes matter? What is the penetration depth of the Raman probe? I assume it does not have a capacity to probe a single layer. Do the authors think that the charge order forms in 3D and is synchronized between the BEDT-TTF planes/ - I assume in any other case they would not observe such a distinct picture, because they would observe different overlapping responses from different layers within penetration depth. These responses would be asynchronized resulting in an averaged picture.

[Reply #2-6]

First, let us note that the Avrami analysis is only available at low temperatures where the examined area contains a number of nucleations. In Methods, we describe the estimate of the penetration depth of the Raman probe for the present material. It is about 1 μm (corresponding to 500 layers) and thus the observed spectrum reflects an average of the charge states over this depth. As seen in Fig. 3a, the Raman image for crystallization from SCL at high temperatures is roughly in blue or red with their boundary graded, meaning that the crystal growth proceeds in the 3D manner probably with a layer-by-layer mixing of SCL and CO only in the boundary region. In contrast, the

gradual colour change in the entire area in Fig.3b cannot distinguish whether individual micro grains grow in a 2D or 3D manner. Since the Avrami exponent suggests growth in 2D, it appears that the seeds extend within each layer with time and get mutually touched in the layers, and overlapped out of the layers. Unfortunately, the Raman probe is unable to distinguish the coherence/incoherence of CO papers in and out of the layers. This issue, namely, how the CO state growing from SCL is similar to that growing from CG regarding coherence, is quite intriguing. In the revised MS, we added the following sentences:

[9th paragraph] “In the present experiments, the laser light penetrates the sample to a depth of about 1 μm which includes 500 layers (see Methods); so, the observed spectrum reflects an average of the charge states over this depth. The Raman image in Fig.3a is roughly in either blue or red with only their boundary graded, meaning that the crystal growth from SCL proceeds in a three-dimensional manner with a possible layer-by-layer mixing of SCL and CO in the boundary region.”

and

[9th paragraph] “This means that the spatial scale of nucleation and growth is considerably smaller than the resolution of 6.5 μm . In this case, the results are unable to distinguish whether the growth is two- or three-dimensional; however, the two-dimensional growth suggested by the Avrami analysis means that nucleated seeds extend with time within each layer to get mutually touched in the layers and may overlap incoherently out of the layers. The in-plane and out-of-plane coherence in the CO growing from CG is an intriguing issue to be investigated.”

[Comment #2-7]

7. Figure 3 shows the snapshots of crystallization. How long does it take to obtain such a snapshot? Is the time shown on the figure a starting time? How does the time of the snapshot acquisition compare to the crystallization rate?

[Reply #2-7]

The data acquisition time for one complete imaging was 149 s for Fig. 3a and 60 s for Fig. 3b. In the case of Fig.3, during this time, crystallization proceeded by 6 μm , which is roughly equal to the spatial resolution of 6.5 μm . This estimate is described in Supplementary VI. The times quoted in Fig. 3 are the starting time of the imaging. We added the following explanation in the caption of Fig.3 in the revised MS.

[Caption of Fig. 3] “..... at 922 s, 2710 s, 3008 s, 3306 s, 3753 s, 4200 s, 4647 s, and 5541 s for the starting time of the imaging. During this imaging, crystallization proceeds by $\sim 6 \mu\text{m}$, which is roughly equal to the spatial resolution of 6.5 μm (see Supplementary Note IV)”

[Comment #2-8]

8. A question about the analysis of crystallization rate from Raman images. Does the

length of the interface between SCL and CO influenced crystallisation? Fig. 3a shows only one seed in the Raman image. Is this picture an example of crystallization images? Do they have any statistics on the crystallization process? I do not see a reason why there should be only one seed in the window of the snapshot. The manuscript tells that it takes time to get the seeds of crystallization in the SCL state. Is the density of those seeds a random number?

[Reply #2-8]

Crystallization from SCL in Fig.3a is a stochastic process, in which nucleation needs a long incubation time but, once a seed nucleates, it rapidly grows. In Fig. S4a (Supplementary Note III), we show Raman images for a larger area, $124.8 \times 416 \mu\text{m}^2$, than shown in Fig.3a; nevertheless, only one seed nucleates and extends over the area before another nucleation occurs. This means that, in a window of the submillimeter scale, multiple nucleations are a rare stochastic case; of course, the probability of multiple nucleations should increase as the window is enlarged. In the revised MS, we added the following explanation:

[9th paragraph] “However, once a part of the SCL crystallizes to form a CO microdomain, it rapidly extends over the entire area *before another nucleation occurs. Even when the observation area is extended to $124.8 \times 416 \mu\text{m}^2$, we observed only one nucleation that extends over the area [Supplementary Note III], meaning that multiple nucleations are a rare stochastic event in the submillimeter scale.*”

Concerning the relation between the length of the SCL/CO interface and crystallization speed, we cannot provide firm information from the data at hand. Nevertheless, we note that the linearity in the inset in Fig. 4a may imply an invariant velocity during crystallization; namely, the length of the interface that increases with time may not affect the crystallization speed.

[Comment #2-9]

9. The authors show a discrepancy of the growth rate of a crystal between a classical SCL and charge SCL they study. Their statement is based on the application of the Wilson-Frenkel model, where they use charge fluctuations rate instead of diffusion coefficient as is done in case of classical crystallization. It is not completely clear to me why the charge fluctuations rate is a good parameter to substitute for the diffusion coefficient. The authors mention earlier in the paper as well as after immediately after the discussion of Wilson-Frenkel model, that the glass is conducting. Therefore there is some charge carriers diffusion associated with the carriers mobility. Can this number be calculated and used as a diffusion coefficient? - it will basically give an qualitative estimate of "electronic orbitals spread" the authors mention. The authors should discuss how this influences crystallisation.

[Reply #2-9]

There is a definite reasons why we used the charge fluctuation rate obtained from the

noise spectrum instead of the diffusion constant of the electron transport. The charge glass state is an interaction-driven collective state with inhomogeneous charge density and importantly has a small energy gap (300 K) to single particle excitations as revealed by the previous transport experiments (Ref.7). Thus, the system shows substantial electrical conductivity by thermally excited single particles (or quasi-particles), which have no direct connection to the motions of CG or SCL state. (The existence of single (quasi-)particle excitations in CG and SCL states is a purely quantum effect not seen in conventional glasses and supercooled liquids.) The electrons travel on the inhomogeneous potential formed by CG or SCL. Then, the fluctuations of CG or SCL act as potential fluctuations for single particle transport and manifest themselves as the noise of resistivity. Thus, instead of the diffusion constant of the electron transport, its noise is informative to the dynamics of CG or SCL, specifically, giving the characteristic frequency of the fluctuations in question. The diffusion constant in the Wilson-Frenkel model generally means the frequency of the CG/SCL fluctuations multiplied by the single step. So, we used the characteristic frequency of the noise [Ref.6] in the Wilson-Frenkel model. To state the above, we added the following sentences in the revised MS.

[12th paragraph] “*We note that the diffusion constant in the Wilson-Frenkel model is defined for the motion that freezes to the glass state upon cooling whereas the diffusion constant derived from the conductivity is for the transport of thermally excited single particles over a small charge gap and has no direct connection to the diffusion constant in question. Then, we used the characteristic frequency of the resistivity noise, $f(T)$, which probes the charge fluctuations responsible for SCL (Ref. 6).*”

[Replies to Reviewer #3]

[Comment #3-0]

The authors have performed Raman microspectroscopy to study crystallization dynamics of electrons in an organic glassy system θ -RbZn.

Raman microspectroscopy allows them to detect a distribution of charge occupancy of the molecular sites, thus distinguishing between an inhomogeneous distribution of charge and the stripe charge ordered phase.

To observe the relevant timescales, the sample was rapidly cooled to a “quench temperature” T_q after which the dynamics of charge ordering was measured. They observed a typical feature that nucleation goes faster as the temperature is lowered, up to a ‘nose temperature’ of $T=165$ K, below which the nucleation of the crystal becomes progressively slower.

By showing the spatial spread of the crystalline phase, they observe two distinct regimes: one with clear nucleation and fast domain growth; whilst at lower temperatures the crystalline nucleation is more homogeneous without domain growth.

At lower temperatures a comparison to the Wilson-Frenkel model fails to explain their data. This, as the authors claim, is due to the “quantum” nature of the glassy freezing.

I think the data presented is really exciting and provides an almost unique and unprecedented insight into glassy slowdown of dynamics. However, the claim that the larger growth rate at low temperatures is attributable to “the quantum effect” is not supported by any clear theory. I can recommend publication of the manuscript, once the authors clarify this statement. That is: either substantiate the “quantum” claim or just stick with the observation that the growth rate violates Wilson-Frenkel theory.

[Reply #3-0]

We are grateful to Reviewer #3 for reading our manuscript thoroughly, highly evaluating it, and giving enlightening comments. In light of the comments, we scrutinized the previous MS and then revised it. Below, we reply to all of the comments and describe the revised parts, which are underlined in the following replies and highlighted by red characters in the MS.

As for the reviewer’s caution on our claim, “the quantum effect”, we agree that our claim should be made on the violation of the classical Wilson-Frenkel model; however, let us mention that the very recent theory of charge glass (Fratini *et al.*, arXiv:2208.06260) has shown that the quantum nature of electrons affects the physical properties of charge glass, then one can expect so in the growth rate. So, we revised the MS to only mention the quantum effect as a possible origin of this violation. So, we rewrote the relevant parts in the MS as follows:

In the abstract, we deleted the underlined parts in the following sentence: “Remarkably, however, the growth rate is many orders of magnitude larger than that in the conventional classical case, ~~which is attributable to the quantum effect.~~”

In the 14th paragraph, we revised the MS in the underlined manner; “*Thus, the*

experimental growth rates are not reconciled with the conventional models. This discrepancy is a key to the microscopic mechanism of the electronic crystallization to be clarified. As a possible relevance to the extraordinarily fast crystallization, we suggest the quantum nature of electrons,.....

and deleted the underlined part:

“Electronic and He crystallizations share anomalous nonequilibrium dynamics, ~~which are attributable to their quantum nature~~”

In the last paragraph (concluding paragraph), the original sentences,
“On the other hand, it has been found that the liquid (or glass) crystal interface proceeds through space orders of magnitude faster than in classical systems, indicating that the quantum nature of electrons is deeply involved in the anomalously high speed of crystal growth.”

is modified as

“On the other hand, it has been found that the liquid (or glass) crystal interface proceeds through space several orders of magnitude faster than expected by the Wilson-Frenkel model relevant to classical systems. We speculate that the quantum nature of electrons is deeply involved in the anomalously high speed of crystal growth. Indeed a recent theory has shown that the quantum nature of electrons affects the physical properties of charge glass [29].”

[29] Fratini, S. et al. A quantum theory of the nearly frozen charge glass. arXiv:2208.06260

[Comment #3-1]

In addition, I have a few other questions and one typo:

- One possible interpretation put forward in the discussion around Fig 4b is that there is a growing length scale “l” (fig 4c) responsible for the large growth rate. This might be an interesting point, and there is a lot of discussion in the glass literature that is not mentioned here, see for example “Direct Experimental Evidence of a Growing Length Scale Accompanying the Glass Transition”, Berthier et al, Science 310, 5755 (2005). I would appreciate if the claim of a growing length scale in Fig 4c is accompanied by an in-depth discussion of these ideas in related glass-problems.

[Reply #3-1]

We appreciate the reviewer’s letting us know about an interesting paper on the experimental evaluation of the dynamical correlation length. We have been interested in the behavior of the dynamical correlation length of the present system; however, some of the soft-matter people whom we discussed with spoke out about the difficulty of experimentally knowing it. Although the method presented in the paper by Berthier et al may not be straightforwardly applicable to the electron glass, we would like to continue to think of its possible modification for future study. Anyway, the increase of the dynamical correlation length toward the glass transition is interesting; so, we mentioned it in the revised MS. However, the value at the glass transition is only several

times the distance between the constituent elements whereas the present value of l is several orders of magnitude larger than the distance. In the revised MS, we added the following explanation, referring to the paper by Berthier et, as follows;

[13th paragraph] “The l may be related to the dynamical correlation length, which was suggested to increase toward the glass transition in colloidal systems [37]. The revealed dynamical correlation length at the glass transition is however several times the colloid spacing: the length scale is quite different from the present case in which l is orders of magnitude larger than the electron spacing.”

[37] Berthier, L. et al. Science **310**, 5755 (2005)].

[Comment #3-2]

- Is there a distinction in Raman signal between the high-temperature phase (electron liquid) and the frozen electron charge glass at low T? Because in both cases, the charge distribution on a single site has a broad distribution, but in the high T case its fluctuating whereas at low T it's freezing.

[Reply #3-2]

The Raman spectrum is somewhat broadened with a small change in its shape on cooling. However, the change is gradual and there is no clear distinction between the spectra in the supercooled electron liquid and electron glass. The time scale of Raman spectroscopy (40 THz in the present case) is much faster than fluctuations responsible for glass formation (slower than 10 kHz) and even faster than thermal fluctuations (1.6-4.2 THz for 80-200 K), so the observed spectrum reflects the snapshot (not motionally narrowed) of the inhomogeneous charge distribution, which is not qualitatively different between in the glass and supercooled states. The gradual temperature dependence of the spectral shape and width is ascribable to thermally activated charge configurations that depend on temperature. For the past three years, we had investigated this issue by extending our scope to a wider range of materials with different lattice frustrations. For details, please see Murase et al., arXiv: 2205.10795, which is referred to in the revised MS. Since we had a related question from Reviewer#2 as well (please see Comment #2-1 and Reply #2-1), we added the following paragraph to jointly reply to Comments #2-1 and #3-2:

[6th paragraph] “Thus, the conversion from CG/SCL to CO is associated with not only charge rearrangement but also a change in charge density profile, distinct from the conventional crystallization of atoms, molecules, and colloids. In particular, the continuous charge density distribution in CG/SCL state (Fig. 2a and Fig. S1) is a remarkable feature of the electronic glassformer. We note that the time scale of Raman spectroscopy, ~45 THz in the present case, is much faster than that of fluctuations responsible for glass formation (slower than 10 kHz) and even faster than thermal fluctuations (1.6-4.2 THz for 80-200 K); so, the observed spectrum projects the snapshot of the inhomogeneous charge distribution not motionally narrowed. Then, the continuous charge density distribution in CG/SC state is argued to come from the frustration-

enhanced quantum-mechanical configuration interaction of various charge patterns and the temperature variation of the distribution width with no clear distinction between CG and SCL states is not due to classical motional narrowing but to thermally activated charge configurations that depend on temperature [28]. Indeed, very recent theoretical work [29] succeeded in explaining these properties of charge glass through quantum simulations.”

[28] Murase et al., Observation of classical to quantum crossover in electron glass. arXiv:2205.10795

[29] Fratini et al. A quantum theory of the nearly frozen charge glass. arXiv:2208.06260

with an addition of a sentence,

“The spectral shape in SCL/CG is gradually and slightly temperature-dependent (Fig. S1).”

[Comment #3-3]

- Typo, in line 114 and 192 there is written “spaciotemporal” instead of spatiotemporal.

[Reply #3-3]

We appreciate the reviewer’s spotting the typo. We corrected them.

REVIEWER COMMENTS

Reviewer #1 (Remarks to the Author):

In their reply ("1-1), they said

"In the present system, f is directly known from the resistivity fluctuations and aging behavior. [Please see also Reply #2-9.] Thus deduced f values were confirmed to be those of the SCL dynamics in that the f values in the resistivity fluctuations and aging behavior are on a single activation function and the glass transition temperature suggested by the temperature dependence of the f values nearly coincide with the practical glass transition observed, e.g., by X-ray diffraction (Refs. 6, 15)."

However, this does not respond to our below criticism in our previous report at all. They just repeat what they describe in their manuscript.

"They claim that the value of l in eq (6) is much larger (10^5 times) larger than the expected value (= lattice constant) in their analysis within this model. According to this fact accompanied with additional considerations, they conclude that their observations reflect the quantum nature of the electron crystallization. However, it does not rule out the possibilities that the above discrepancies of " l " may be understood by a different mechanism.

In eq. (6), they use the frequency of the fluctuation obtained from the noise spectroscopy. This frequency would inevitably become low because the observed fluctuations are limited by the macroscopic aspects of the fluctuation, even if electronic fluctuations exist. The authors should use a more macroscopic crystal growth step accounting for the macroscopic fluctuation (instead of the microscopic one they employ such as the lattice constant). The discrepancy of 10^5 might almost disappear, if they use a macroscopic step in their analysis. In this manuscript and its references, I could not find any justification for employing the microscopic growth step for the slow fluctuations. From this standpoint, the conclusion of this manuscript is not justified."

I also could not find any additional justification in Reply #2-9 and the references. Thus, they just repeat their opinion in the previous reply. Therefore, I regretfully cannot recommend this manuscript for publication. But, I will not obstruct the publication, if other reviews give positive comments for the publication. (I have already written positive features of their work in the previous report.)

Reviewer #2 (Remarks to the Author):

I appreciate the authors answering all the comments I provided. I am also satisfied with the replies the authors provided to the comments of the other two Reviewers. I find that manuscript has been improved, and the data on the electronic crystallization are of high interest. Therefore I recommend the manuscript for publication.

Reviewer #3 (Remarks to the Author):

My questions and concerns have been addressed, I can recommend publication without further changes.

Reply to Reviewer #1's comment

[Comment]

In their reply ("1-1), they said

"In the present system, f is directly known from the resistivity fluctuations and aging behavior. [Please see also Reply #2-9.] Thus deduced f values were confirmed to be those of the SCL dynamics in that the f values in the resistivity fluctuations and aging behavior are on a single activation function and the glass transition temperature suggested by the temperature dependence of the f values nearly coincide with the practical glass transition observed, e.g., by X-ray diffraction (Refs. 6, 15)."

However, this does not respond to our below criticism in our previous report at all. They just repeat what they describe in their manuscript.

"They claim that the value of l in eq (6) is much larger (10^5 times) larger than the expected value (= lattice constant) in their analysis within this model. According to this fact accompanied with additional considerations, they conclude that their observations reflect the quantum nature of the electron crystallization. However, it does not rule out the possibilities that the above discrepancies of " l " may be understood by a different mechanism.

In eq. (6), they use the frequency of the fluctuation obtained from the noise spectroscopy. This frequency would inevitably become low because the observed fluctuations are limited by the macroscopic aspects of the fluctuation, even if electronic fluctuations exist. The authors should use a more macroscopic crystal growth step accounting for the macroscopic fluctuation (instead of the microscopic one they employ such as the lattice constant). The discrepancy of 10^5 might almost disappear, if they use a macroscopic step in their analysis. In this manuscript and its references, I could not find any justification for employing the microscopic growth step for the slow fluctuations. From this standpoint, the conclusion of this manuscript is not justified."

[Reply]

First, let us note that Reviewer's critique is based on a misconception of resistivity fluctuations. He/she mentioned that "*This frequency would inevitably become low because the observed fluctuations are limited by the macroscopic aspects of the fluctuation, even if electronic fluctuations exist.*". The resistivity fluctuations originate

from the microscopic scattering of electrons. Even if the characteristic frequency of the resistivity noise is low, the noise originates in microscopic origins.

Second, Reviewer mentioned that *“The authors should use a more macroscopic crystal growth step accounting for the macroscopic fluctuation (instead of the microscopic one they employ such as the lattice constant). The discrepancy of 10^5 might almost disappear if they use a macroscopic step in their analysis.”* This statement is hard for us to decipher. As we explained in our manuscript and also emphasized in our previous reply, crystals grow step by step with constituent particles stacking on the crystal surfaces. In the conventional model, the “step” corresponds to the distance of one or several particle sizes, which are equivalent to the order of lattice spacing in the present case, as we described in the manuscript. The present work finds that the step is orders of magnitude larger than the lattice spacing. This macroscopic step deduced in the present study is surprising and it is the point of the present work. Additionally, Reviewer says, “I could not find any justification for employing the microscopic growth step for the slow fluctuations.” **Let us emphasize again that the microscopic growth step is the conventional wisdom of crystal growth, and the present consequence of the macroscopic growth step is surprising.**

[Comment]

I also could not find any additional justification in Reply #2-9 and the references. Thus, they just repeat their opinion in the previous reply. Therefore, I regretfully cannot recommend this manuscript for publication. But, I will not obstruct the publication, if other reviews give positive comments for the publication. (I have already written positive features of their work in the previous report.)

[Reply]

As we explained above, we think Reviewer#1's comments come from his/her misconception of the resistivity noise.

We discussed the possibility of a further revision of our manuscript. However, we have decided not to make a further revision because we are afraid that it would make the manuscript too tedious for general readers.

Finally, we thank Reviewer#1 for his/her evaluation of the positive feature of our work and his/her statement, *“But, I will not obstruct the publication, if other reviews give positive comments for the publication”*. Let us note the Reviewer#2’s statement, *“I am also satisfied with the replies the authors provided to the comments of the other two Reviewers.”*